# DR-CFGNN: A Completion-Aware Framework for Counterfactual Explainability in Graph Neural Networks

## Abstract

In this study, we propose a novel framework for counterfactual explainability in graph neural networks (GNNs). To the best of our knowledge, this is the first generic, model-agnostic method for local-level GNN explainability that considers both edge removal and edge assertion. The approach takes advantage of the progress achieved in factual explainability, coupling it with an encoder-decoder deep learning model to learn valid and robust graph expansions. In addition to standard benchmark datasets, we evaluate our method on a new variant of a popular synthetic dataset to study how explainability is influenced by data incompleteness, a common characteristic of real-world graph data. A multi-faceted experimental analysis with both established metrics from relevant literature and novel ones aimed at assessing the validity and the quality of explanations, demonstrates the advancement that our proposed approach brings to state-of-the-art baselines.

## 1 Introduction

The generation of counterfactual explanations (CFXs) is often advocated as a highly prominent direction for tackling the lacking interpretability of black box, data-driven Artificial Intelligence (AI) models (Jiang et al., 2024a; Guo et al., 2025; Kaddour et al., 2025). Unlike factual explanations that aim to identify a mapping from the input and model features to the outcome, thus revealing aspects that seem correlated with the predictions of the model, CFXs explore the impact of parameters that, if modified, can drive the model to alter the original prediction. Such explanations are closer to human intuition and offer the means for recourse, facilitating deliberation on which actions to perform to reach a desirable outcome (Miller, 2019).

Driven by the pervasiveness of graph-structured data across a plethora of domains, the demand for reliable and trustworthy AI systems has grown lately to include also Graph Neural Networks (GNNs). By performing representation learning over graph data, GNNs successfully accomplish predictive tasks in a wide range of diverse fields, from biological (Wieder et al., 2020) and medical (Geng et al., 2024) networks in life sciences, e.g., for drug discovery (Besharatifard & Vafaee, 2024), to the identification of material properties in materials science and engineering (Shi et al., 2024), up to trading networks in the financial industry (Gao et al., 2024; Lv et al., 2019), social networks (Wang et al., 2023), transportation networks (Jiang et al., 2024b), and more.

For GNNs, explainability and interpretation have proven more challenging compared to other neural models (Yuan et al., 2023). Most relevant research has concentrated on local-level, post-hoc factual explainers (Dai et al., 2024; Yuan et al., 2023; Agarwal et al., 2023), which, given a trained GNN model and one or more input graphs, aim to extract a subgraph of the input that plays crucial role for the target model to generate a specific prediction. Only recently counterfactual explainers started to gain traction (Guo et al., 2025), with initial attempts borrowing insights from the progress achieved in factual GNN explainability, applying various techniques for edge and feature masking.

Despite their effectiveness, the current state of counterfactual explainability for GNNs is marked by notable limitations: (a) The vast majority of local-level approaches (see Section 2.2) do not explore graph expansions at all; they limit the search for counterfactuals to subgraphs of the input graph, in a style similar to factual explainers. Edge additions though can significantly broaden the informativeness and applicability of counterfactuals, especially in the case of incomplete data. In

fact, incomplete graphs are prevalent in a wide spectrum of real-world domains (Xia et al., 2025; Arrar et al., 2024; Chen et al., 2022), constituting hypotheses in the form "*should node $x$ was connected to node $y$, then ...*" a crucial means of recourse. (b) There is a handful of local-level GNN explainers that consider graph expansion, but this is usually accomplished by following domain-specific methods, e.g., LEGIT (Bacciu & Numeroso, 2023) and MEG (Numeroso & Bacciu, 2021) that apply Reinforcement Learning in order to exploit domain knowledge for molecular applications, or by applying assumptions that are difficult to generalize, e.g., GREASE (Chen et al., 2025), which focuses on recommender systems, and CLEAR (Ma et al., 2022), which assumes an underlying causal model (c) Graph expansion has also been considered by certain global-level explainers, which aim to identify high-level, global rules that apply to large proportions of the input graphs, e.g,. GCFExplainer (Huang et al., 2023b). The global perspective is indeed valuable, especially for factual explainability; we believe though that one of the primary objectives of generating CFXs, i.e., to reveal the well-targeted, minimal graph edits that lead to a desirable outcome under certain criteria, is better served by local explainability, where guidelines are provided for each individual input graph separately. (d) Finally, most models implement costly search-based methods or run-time perturbations and random-walks for each individual graph provided in the input.

In this study, we concentrate on the problem of model-agnostic, local-level, post-hoc CFXs of GNNs, focusing, but not being restricted, to the task of graph classification. We make the following contributions: First, we formalize and implement the DR-CFGNN (Deconstruction-Reconstruction Counterfactual GNN) explainer, a general-purpose, novel framework for counterfactual explainability of GNNs. Unlike prior methods, DR-CFGNN treats edge addition and edge removal independently rather than end-to-end, exploiting more effectively the implications they impose on the graph.

Second, in addition to established metrics used in relevant literature that mainly focus on the ability of explainers to generated CFXs, we assess the quality of explanations, too. We expand the notion of robustness of GNN explanability, bringing it closer to the practice already adopted in other areas of Machine Learning. We further introduce more fine-grained metrics to assess explanation quality.

Third, we conduct a multi-faceted experimental evaluation with synthetic and real-world datasets, demonstrating the superior performance of the proposed approach. More importantly, we reveal for the first time the impact of counterfactuals in dealing with incomplete graph data: we suggest a novel variant of a popular benchmark and demonstrate the efficacy of the completion-aware approach of DR-CFGNN to generate more accurate and intuitive explanations.

## 2 RELATED WORK

### 2.1 FACTUAL EXPLAINABILITY OF GNNS

Compared with deep neural models for images and text, the explainability of GNNs is less explored. Recent review articles draw the picture of the current progress (Yuan et al., 2023; Longa et al., 2025; Dai et al., 2024). Evidently, the majority of GNN explainers are factual, instance-level, post-hoc models. GNNExplainer (Ying et al., 2019), one of the most popular to date models, optimizes soft masks on edges and attributes, aiming to maximize the number of those that can be eliminated while preserving original predictions as much as possible. Similar perturbation techniques are employed by other explainers too, including SubgraphX (Yuan et al., 2021) that performs Monte Carlo tree search to find the most important subgraph, ZORRO (Funke et al., 2021) that uses fidelity to revise the search process, or PGExplainer (Luo et al., 2020) and GraphMask (Schlichtkrull et al., 2021) that train a deep neural network to predict effective edge perturbation masks. The emphasis on generating subgraphs is shared among all these models. GraphLime (Huang et al., 2023a) on the other hand, employs a surrogate model which can assign large weights to features that are important.

Recently, self-explainable GNNs, such as SE-GNN (Dai & Wang, 2021) and SES (Huang et al., 2024), have been proposed that provide explanations in parallel with predictions. The progress in factual GNN explainability is rapid, creating high expectations for future advancement.

### 2.2 COUNTERFACTUAL EXPLAINABILITY OF GNNS

Contrary to factual GNN explainers, counterfactual learning on graphs has a very short history, as detailed in recent survey articles (Guo et al., 2025; Kaddour et al., 2025). Building off of the momen-

tum gained in factual explainability, most approaches target at local-level, post-hoc explanations. For instance, CF-GNNExplainer (Lucic et al., 2022) iteratively optimizes a binary perturbation matrix, while RCExplainer (Bajaj et al., 2021) learns the shared decision region of the target GNN across multiple input graphs and generates edge masks trying to optimize that the input graph and the counterfactual graph reside on the opposite side of the decision boundary. MOO (Liu et al., 2021) and $CF^2$ (Tan et al., 2022) balance the learning of masks between optimizing the factual and counterfactual explanation generation, with heuristically determined search algorithms. CFExplainer (Chu et al., 2024), on the other hand, optimizes the perturbation of the mask in a way that reduces the likelihood of producing the original prediction. These models adapt effectively the different strategies to the requirements of counterfactual explainability, but, as already mentioned, they only consider edge removal, leading to explanations that constitute subgraphs of the input graph.

Edge addition has been incorporated in LEGIT (Bacciu & Numeroso, 2023) and MEG (Numeroso & Bacciu, 2021), highlighting the benefits of generating CFXs. Yet, the specification of all possible states needed by their Reinforcement Learning module requires extensive domain knowledge, making it difficult to generalize the approach. CLEAR (Ma et al., 2022) adopts a different approach, based on an encoder/decoder architecture whose regularization term enforces the model to make minimal changes to the graph structure. This direction, similar to our proposed model, adds the extra benefit of learning how to transform the graph, rather than working iteratively with every single input as the previous models, significantly improving efficiency and generalizability. From a rather similar standpoint, GREASE (Chen et al., 2025) adopts a Relational Graph Convolutional Architecture to learn a surrogate model to generate CFXs. Yet, both models learn edge removal and addition end-to-end, which, as we show in the sequel, can be suboptimal in the general case. Moreover, GREASE is domain-specific, particularly designed for recommendation tasks, while CLEAR requires the existence of an underlying causal model of the data, which often is not available.

Recently, the goal of generating CFXs that apply to big proportions of the input graphs has been adopted by global counterfactual explainers, such as GCFExplainer (Huang et al., 2023b) and GlobalGCE (He et al., 2025). Despite the different focus, we again observe the same pattern: the former model both adds and removes edges, but adopts a random walks approach, while the latter, aiming to reduce complexity, implements an autoencoder similar to ours, in order to generate important counterfactual graphs, yet concentrates on subgraph explanations only. The novel design of DR-CFGNN offers a generalization that can influence state-of-the-art models in the field.

## 3 PROBLEM FORMULATION

We denote by $\Phi : \mathcal{G} \to \mathcal{Y}$ a trained GNN classifier, where $\mathcal{G}$ denotes a set of homogeneous, undirected graphs and $\mathcal{Y}$ is the space of labels. Let $G = \{V, E\} = \{\mathbf{A}, \mathbf{X}\}$ two alternative representations of a graph $G \in \mathcal{G}$, where $V = \{v_1, .., v_N\}$ the set of nodes, $E \subseteq V \times V$ the set of edges, $\mathbf{A} \in \{0, 1\}^{N \times N}$ the binary adjacency matrix , and $\mathbf{X} \in \mathbf{R}^{N \times d}$ the feature matrix.

For a given prediction $\Phi(G) = y$, a factual GNN explainer $\Psi_{\Phi, F}(\cdot)$ aims to find one or more factual graphs $G^F \subseteq G$ to explain the prediction of $\Phi$, s.t. the prediction label of the subgraph does not change, i.e., $\Phi(G^F) = \Phi(G)$. Similarly, a counterfactual explainer $\Psi_{\Phi, CF}(\cdot)$ aims to find one or more counterfactual graphs $G^{CF}$, which have minimal and *reasonable* changes to $G$ and lead to a different prediction label, i.e., $\Phi(G^{CF}) \neq \Phi(G)$.

Staying close to the generic representation of Guo et al. (2025), let also $\overline{\mathbf{A}} = \mathbf{J} - \mathbf{I} - \mathbf{A}$ be the supplement of the adjacency matrix $\mathbf{A}$ of a graph $G$, where $\mathbf{J}$ is an all-one matrix and $\mathbf{I}$ an identity matrix. Then, let $\mathbf{C} = \overline{\mathbf{A}} - \mathbf{A}$ be the matrix storing all possible edge perturbation candidates that can be performed on $\mathbf{A}$. Specifically, $\mathbf{C}$ can be decomposed as $\mathbf{C} = \mathbf{C}^+ + \mathbf{C}^-$, where $\mathbf{C}^+ \in \{0, 1\}^{N \times N}$ and $\mathbf{C}^- \in \{-1, 0\}^{N \times N}$. Positive elements in $\mathbf{C}^+$ (resp. negative elements in $\mathbf{C}^-$) denote edges in $\mathbf{A}$ that do not exist and can potentially be added (resp. edges that exist and can be removed).

The search for finding counterfactual graphs can benefit from both adding edges, e.g., when the data are incomplete, and deleting edges. To determine where on the input graph to make the changes is an area of open research, with approaches ranging from random walks and perturbations to exhaustive heuristic-based searching and domain-specific algorithms. A generic representation of the problem can use mask notation, placing masks over $\mathbf{C}$ to specify the desirable changes on the input graph:

$$\mathbf{A}' = \mathbf{A} + \mathbf{C}^+ \odot \mathbf{M}^+ + \mathbf{C}^- \odot \mathbf{M}^-, \tag{1}$$

where $\odot$ denotes element-wise product and $\mathbf{M}^+, \mathbf{M}^- \in \{0, 1\}^{N \times N}$ are masks that select from all possible candidate operations the ones judged most appropriate to form a new graph. Similarly, a mask $\mathbf{F}$ can be applied on the feature space $\mathbf{X}$ to hide features not affecting the given prediction:

$$\mathbf{X}' = \mathbf{X} \odot \mathbf{F} \tag{2}$$

As a result, the problem of finding CFXs can be converted to an optimization problem as follows:

**Definition 3.1** (*Counterfactual Explanation Problem*). Given a GNN $\Phi : \mathcal{G} \rightarrow \mathcal{Y}$, an input graph $G \in \mathcal{G}$ with $G = \{\mathbf{A}, \mathbf{X}\}$ and a prediction $\Phi(G) = y$ with $y \in \mathcal{Y}$, the CFX problem is defined as the optimization problem of finding proper masks $\mathbf{M} = [\mathbf{M}^+, \mathbf{M}^-]$ and $\mathbf{F}$ that, with the help of Eq. 1 and 2, generate a new graph $G^{CF} = \{\mathbf{M}, \mathbf{F}\}$, such that $\Phi(G) \neq \Phi(G^{CF})$, under certain criteria.

In relevant literature, the most common criterion applied to Definition 3.1 is sparsity, i.e., the less graph edits possible, but others are often discussed, such as maximal discrepancy between the factual output and the counterfactual output.

## 4 THE PROPOSED FRAMEWORK: THE DR-CFGNN EXPLAINER

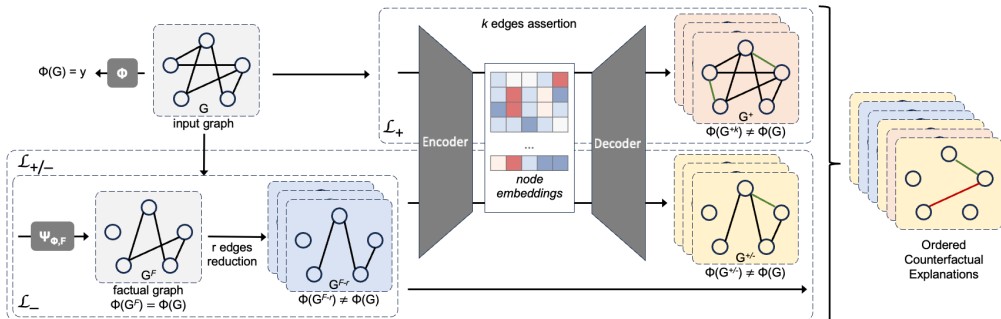

Figure 1: Illustration of the DR-CFGNN framework.

As discussed in Section 2, the majority of current explainers generate CFXs that are subgraphs of the input graph, i.e., they only focus on optimizing $\mathbf{M}^-$ in Eq. 1, or train the $\mathbf{M}^-$ and $\mathbf{M}^+$ masks end-to-end. We argue in this study that a counterfactual explainer can benefit by distinguishing between the task of determining where to add edges from the task of determining which edges to remove. We propose the DR-CFGNN framework, a new, unified model, where the process of CFX generation is broken down into separate tasks (Figure 1):

- **Deconstruction.** Primarily through the deletion of edges, this step does not explore the complete space of candidates, but instead focuses exactly on that part of the input graph that drives the target GNN to generate its original prediction, i.e., the factual subgraph.
- **Reconstruction.** Aiming at generating a different prediction label, this step focuses on completing partial patterns that may exist in the graph, learned from the same domain-specific knowledge as used to train the target GNN.

The intuition is that the deconstruction operation ruins the patterns responsible for a given prediction. This part can benefit from the rapid progress observed in factual GNN explainability. Reconstruction takes advantage of domain knowledge learned during training, e.g., valid molecule structures, stable chemical bonds etc., resembling the process of link prediction. This design can produce CFXs in a more accurate and efficient manner, as the two operations work complementary to one another, driven by the characteristics of the data.

### 4.1 DECONSTRUCTION

Given the problem definition 3.1, we formalize this step by defining the loss function $\mathcal{L}_-$ to optimize the edge removal process, as follows:

$$\mathcal{L}_- = \mathbf{A} + \mathbf{C}^- \odot (\mathbf{A}^F - \mathbf{A}^{F-r}), \tag{3}$$

where we rely on $\Psi_{\Phi,F}(\cdot)$, a typical factual explainer, to provide a factual subgraph $G^F = \{\mathbf{A}^F, \mathbf{X}^F\}$ of size $N' << N$, so that $G^F \subset G$ and $\Phi(G^F) = \Phi(G)$. Given $G^F$, we obtain the set $\mathcal{G}^{F-r}$ of *all* subgraphs of $G^F$ from which $r$ edges have been removed. Considering that both $N'$ and $r$ are small, the number of $G^{F-r} = \{\mathbf{A}^{F-r}, \mathbf{X}^F\}$ graphs generated is manageable. This process can produce multiple counterfactuals for each input graph; the optimization of $\mathcal{L}_-$ aims to find the minimum $r$, such that the prediction of the target GNN changes. Note that the mask $\mathbf{M}^- = (\mathbf{A}^F - \mathbf{A}^{F-r})$ generated is applied on the input graph, i.e., we eventually remove $r$ edges from the input graph, aiming to minimally deconstruct the pattern that led to the original prediction.

Our current implementation relies on SubgraphX (Yuan et al., 2021) to generate the factual subgraph, since it significantly outperforms other factual explainers in the literature (Yuan et al., 2023; Serra & Niepert, 2022); however, this step is not bound to a particular explainer. SubgraphX uses Monte Carlo Tree Search to explore candidate subgraphs efficiently and evaluates them with Shapley values. Shapley values quantify each subgraph's contribution to the prediction while accounting for interactions with all other nodes or subgraphs, rather than relying solely on raw prediction scores.

## 4.2 Reconstruction

The reconstruction step aims to find modifications to the input graph that create motifs or structures that lead the target GNN to a different prediction. To achieve this, we implement a GNN network $\Psi^+(\cdot)$ that takes as input a graph $G$ and proposes a set of additional edges, which, when unified with $G$, cause the original prediction to change. We define the loss function :

$$\mathcal{L}_+ = \mathbf{A} + \mathbf{C}^+ \odot \mathbf{A}^{+k} \tag{4}$$

that finds graphs $G^{+k} = \{\mathbf{A}^{+k}, \mathbf{X}\}$ having up to $k$ new edges (a mask on the features is not relevant for graph expansion, at least not for homogeneous graphs).

To accomplish this step, we introduce a graph convolutional encoder–decoder deep learning model that outputs the likelihood of previously non-existent edges between node pairs. This allows us to add edges in graphs when generating counterfactuals. Most existing research focuses on predicting missing edges in a single graph (Wu et al., 2022; Li et al., 2023; Yun et al., 2021). In contrast, our approach extends the standard learning-based link prediction setting (Zhang & Chen, 2018) to a dataset containing many graphs. This setting enables the encoder–decoder to capture structural and feature-based patterns that repeat across graphs and to learn general principles of edge formation.

Formally, given the graph $G = \{A, X\}$, the encoder embeds node features into a latent space by iteratively propagating messages over the adjacency matrix:

$$\mathbf{H}^{(\ell+1)} = \sigma\left(\hat{A}\,\mathbf{H}^{(\ell)}\mathbf{W}^{(\ell)}\right), \quad \mathbf{H}^{(0)} = X,$$

where $\hat{A}$ is the normalized adjacency matrix with self-loops, $\mathbf{W}^{(\ell)}$ are learnable weights, and $\sigma(\cdot)$ is a non-linear activation (Kipf & Welling, 2017). After $L$ layers, the encoder outputs final node embeddings $\mathbf{Z} = \mathbf{H}^{(L)}$. The decoder then predicts edge existence by concatenating the embeddings of two nodes $u, v \in V$ and passing them through a multi-layer perceptron $f_\theta$:

$$\hat{y}_{uv} = \sigma\left(f_\theta\big([\mathbf{Z}_u \,\|\, \mathbf{Z}_v]\big)\right),$$

where $[\cdot \,\|\, \cdot]$ denotes concatenation, and $\hat{y}_{uv} \in (0, 1)$ is the likelihood that a $(u, v)$ edge exists.

We train the model with the same dataset used for $\Phi(\cdot)$, further splitting it into training, validation, and test graphs. The test instances of the overall dataset are never used to the edge addition model, as doing so would constitute data leakage. For the training graphs, we further split their edges into two categories: *message passing edges*, used by the encoder to propagate information between nodes, and *supervision edges*, used by the decoder to learn edge existence. The splitting is performed in a disjoint manner, ensuring that the message passing and supervision edges do not overlap. Separate validation or test edges are not included in the training graphs. For validation and test graphs, we apply a similar edge-splitting procedure individually for each graph. During this step, a small fraction of edges is left for evaluation, while the remaining edges are used to pass messages during encoding. Furthermore, edge labels are used: positive edges (labeled as 1) correspond to real connections observed in the graph, while negative edges (labeled as 0) are generated through negative sampling and represent non-existent connections. The decoder is trained to distinguish between these two cases.

Ultimately, during the reconstruction step of DR-CFGNN all nonexistent candidate edges of a graph are fed into the trained encoder-decoder to estimate their likelihood. This design makes our approach efficient and generalizable: the model is trained once, and during the reconstruction step, it only performs inference on unknown test graphs. Unlike other methods that require perturbations or optimization for each individual graph, the approach generates counterfactuals more efficiently.

### 4.3 DE\RE-CONSTRUCTION

Having the two distinct modules that learn where to remove and where to add edges to the input graph, we define also the loss function for both edge editing tasks to the graph, as follows:

$$\mathcal{L}_{+/-} = \mathbf{A} + \mathbf{C}^- \odot (\mathbf{A}^F - \mathbf{A}^{F-r}) + \mathbf{C}^+ \odot \mathbf{A}^{+k} \tag{5}$$

Even though this step relies on the previous modules, it does not subsume them; the counterfactuals produced by each of the three individual steps are distinct (see the Ablation Study in Section 5.6).

### 4.4 UNIFICATION AND ORDERING OF CANDIDATES

While many factual and counterfactual explainers generate a single explanation graph, the three steps of the DR-CFGNN framework (Figure 1) produce different sets of candidate counterfactual graphs. It then depends on the optimization of the loss functions and the desirable additional criteria to define a (partial) ordering among them. Overall, the objective function that the DR-CFGNN framework optimizes, in order to generate counterfactual graphs, is

$$\min_{k,r} \mathcal{L}_{pred} = \mathcal{L}_- + \mathcal{L}_+ + \mathcal{L}_{+/-} \tag{6}$$

The CFX of each input graph is then the edges that need to be added to and/or removed from the input graph, along with the features that are masked out.

## 5 EXPERIMENTS

We evaluate the performance of DR-CFGNN through a series of experiments across various metrics and datasets. The experiments aim to address key aspects relevant to counterfactual explainability: (RQ1) Does our method have sufficient coverage, i.e., can it produce counterfactuals? (RQ2) Are the generated explanations concise and accurate enough to be human-understandable? (RQ3) Are the explanations robust against noise? (RQ4) Are the CFXs produced consistent under noisy conditions? It is noteworthy that due to the challenging nature of the problem and the lack of standardized benchmarks, many counterfactual GNN explainers lay emphasis on RQ1 alone. Implementation details for the GNN and DR-CFGNN models are provided in Appendix A.1.1 and A.1.2, respectively.

### 5.1 DATASETS

We evaluate our proposed framework on real-world datasets of diverse characteristics. Due to the lack of ground-truth explanations, synthetic datasets are also used.

The **BA-2Motifs** (Luo et al., 2020) dataset is a synthetic graph classification dataset consisting of 1000 graphs. Each graph is created by attaching either a five-node cycle motif or a five-node house motif to a base graph generated using the Barabási–Albert (BA) model. The graphs are labeled 0 for the cycle motifs and 1 for house motifs. The node features are of size 10, with all entries set to 0.1.

**Graph-SST2**, **Graph-SST5**, and **Graph-Twitter** (Yuan et al., 2023) are graph classification datasets containing $70,040$, $11,855$, and $6,940$ graphs, respectively, with 2, 5, and 3 classes. They are designed for sentiment analysis tasks, where the highest class corresponds to the most positive sentiment. Each graph represents a sentence, nodes represent words, edges represent relationships between words. Node features, extracted using a pre-trained BERT model, have a size of 768.

**BBBP** (Martins et al., 2012) is a molecular graph classification dataset that consists of $2,050$ graphs. It is used to predict the permeability of Blood-Brain Barrier (BBB). Each compound is represented as a graph, where atoms are represented as nodes and bonds as edges. Each node has a feature vector of size 9 derived from the molecular structure of the compound and each graph is labeled 0 or 1 to indicate blood–brain barrier permeability.

Finally, motivated by the lack of an explicit evaluation of the performance of GNN explainers when dealing with incomplete data, we also construct a variant of the BA-2Motifs dataset, named **BA-2Motifs-3classes**. Specifically, after following the same graph generation process as BA-2Motifs, we randomly remove a single edge from the motif of one third of the input graphs; we assign a third label, namely label 2, to this class of partially complete input graphs.

## 5.2 EVALUATION METRICS

Simplicity is a crucial aspect for the generation of comprehensive explanations, in order to enhance human perception and reaction time (Lynn & Bassett, 2020). Nevertheless, the assessment of the quality of an explanation is less often discussed. Given both needs, we consider the following metrics (their formalization is given in Appendix A.2):

**Probability of Necessity (PN)**: PN (Chu et al., 2024) measures the coverage capacity of an explainer. It computes the proportion of input graphs for which at least one counterfactual exists.

**Explanation Size**: The explanation size is the total number of edge removals and additions that constitute the explanation, along with the number of features masked. For models that generate more than one CFX for a given instance, we apply a partial ordering that prioritizes minimal explanations.

**Motif Proximity**: A quality criterion for a CFX to be intuitive for humans is to be relevant to the input, which in the case of graphs can be seen as the ability to modify edges that adhere to the motif that generates the original prediction. Following Ying et al. (2019) and Luo et al. (2020), we apply a variation of the accuracy metric to measure the proximity of the explanation to the motif, applicable when ground-truth is available. Namely, we introduce Motif Proximity that calculates the proportion of edges in the explanation that are connected to the motif of the input graph; then, among the different counterfactuals produced for each input, the metric takes the best (maximum proximity) and averages this proportion among all instances for which CFXs have been produced.

**Validity after Noise**: Robustness of CFXs, i.e., the validity of explanations under changing conditions, is a major topic in the broader field of AI, but is rarely discussed in the context of counterfactual GNN explainability. According to Jiang et al. (2024a), an important aspect of robustness is related to noise in the input, called Validity after Noise (VaN). Generalizing the notions of Bajaj et al. (2021), which can only by applied to explainers that calculate edge weights, we define VaN as the fraction of counterfactual graphs that remain valid when small noise $\sigma$ is added to the input graphs without changing the prediction. Given that CF explainers manage to generate counterfactuals only for some of the input graphs, this fraction alone can be misleading. In order to constitute the results more dependable, we implement VaN with the help of Wilson Score Interval (Wilson, 1927) that is especially suited for capturing confidence in small sample sizes. We set a confidence level of $95\%$.

**Edge Consistency after Noise (ECaN)**: GNN explainers that do consider robustness are mostly limited to metrics similar to VaN that measure the ability to remain robust against noise; yet, this perspective alone fails to reveal the impact that this noise has on the quality of the explanation. In this respect, we introduce a new perspective, which considers the structural stability of the counterfactual explanations. In particular, we define ECaN that calculates the Jaccard similarity of the counterfactual explanations before and after noisy inputs.

## 5.3 BASELINES

For comparison purposes, we compare the performance of DR-CFGNN against an exhaustive search baseline and CFExplainer (Chu et al., 2024), a state-of-the-art counterfactual explainer particularly designed for graph classification tasks. The **naive random** baseline generates counterfactual graphs by randomly removing or/and adding edges to the input graph. For each graph, we explore all combinations of $r = 0, 1, 2$ edge deletions and $k = 0, 1, 2$ edge additions (excluding the trivial case $r = k = 0$) until a counterfactual is found. Apparently, this approach is computationally expensive and cannot scale to larger graphs, while often producing counterfactuals that are not meaningful, as shown in Section 5.4; yet, it helps set a reference point, against which to contrast the performance of approximate methods. The **CFExplainer** (Chu et al., 2024) on the other hand perturbs the input graph, by optimizing a loss function that aims to reduce the likelihood that the generated graph still produces the original prediction, while keeping the perturbation minimal.

## 5.4 QUANTITATIVE ANALYSIS

Table 1: PN % (Explanation Size)

| Dataset | Naive Random | CFEx1 | CFEx2 | CFEx3 | CFEx4 | CFEx5 | CFEx6 | DR-CFGNN |
|---------|-------------|-------|-------|-------|-------|-------|-------|----------|
| BA-2Motifs | 100 (1) | 24.49 | 37.76 | 43.88 | 50 | 51.02 | 51.02 | 81.63 (1.14) |
| BA-2Motifs-3classes | 100 (1) | 56.1 | 70.73 | 70.73 | 71.95 | 71.95 | 71.95 | 92.68 (1.03) |
| BBBP | 63.1 (1.51) | 5.95 | 10.71 | 14.88 | 19.64 | 23.81 | 32.74 | 29.16 (1.67) |
| Graph SST2 | 28.31 (1.49) | 1.74 | 2.24 | 2.55 | 2.43 | 2.55 | 2.55 | 4.98 (1.54) |
| Twitter | 41.72 (1.39) | 3.48 | 2.66 | 4.09 | 4.91 | 6.13 | 5.11 | 18.2 (1.64) |
| Graph SST5 | 53.85 (1.33) | 8.69 | 10.30 | 11.38 | 11.47 | 12.28 | 12.46 | 24.19 (1.39) |

Table 2: Motif Proximity

| Dataset | Naive Random | CFEx1 | CFEx2 | CFEx3 | CFEx4 | CFEx5 | CFEx6 | DR-CFGNN |
|---------|-------------|-------|-------|-------|-------|-------|-------|----------|
| BA-2Motifs | 0.582 | 0.5000 | 0.5541 | 0.4651 | 0.4949 | 0.4600 | 0.4767 | **0.923** |
| BA-2Motifs-3classes | 0.098 | 0.4348 | 0.3190 | 0.3161 | 0.3051 | 0.3085 | 0.3249 | **0.950** |

**RQ1 & RQ2** Table 1 reports coverage results across datasets and methods, measured by the Probability of Necessity (PN). For the naive random baseline and for DR-CFGNN the explanation size is shown in parentheses (max size up to 4, see Appendix A.1.1), while for CFExplainer, given that the CFXs needs to be fixed before execution, we test 6 different variants of size ranging from 1 to 6.

As expected, the naive random achieves the highest PN values across all datasets (bolded), due to its exhaustive search method, with a 60-second limit per graph. Evidently, DR-CFGNN produces very compact explanations (average size below 1.7 across all datasets) while it still significantly outperforms almost all CFExplainer variants in its capacity to generate CFXs.

This performance can better be appreciated if coupled with the capacity to generate CFXs of high quality. Table 2 investigates this perspective through the Motif Proximity metric, reporting results for the synthetic datasets only, for which ground truth is available. The design choices of DR-CFGNN that assist in learning where to apply graph edits has remarkably elevated its capacity to stay close to important motifs while searching for counterfactuals. Despite the high PN values, the naive baseline fails to capture meaningful explanations.

**RQ3** To evaluate robustness, we perturb the input graphs by adding or removing $4\%$ of their edges and apply Gaussian noise to the features of $4\%$ of the nodes ($\sigma = 0.02$ for synthetic datasets, $\sigma = 0.04 \times$ feature std for sentiment datasets), typically affecting only one node or edge. BBBP was excluded, as adding noise could produce invalid molecular graphs. We also exclude the naive baseline, as its random edge modification process does not help draw valuable conclusions.

Figure 2 shows Validity after Noise (VaN) (Eq. 11) across the different datasets. DR-CFGNN proves more robust than CFExplainer with higher confidence in all but the BA-2Motifs datasets. It becomes evident that CFExplainer needs larger, less compact explanations, in order to improve its robustness. Overall, DR-CFGNN maintains a mean robustness score close to or above $0.6$ across datasets of diverse characteristics. Table 6 in Appendix A.3 lists the raw VaN values.

**RQ4** To further demonstrate the robust nature of DR-CFGNN , we measured the similarity of explanation edges between original and noisy counterfactuals using the Jaccard similarity (Eq. 12). DR-CFGNN significantly outperforms all variants of CFExplainer across all datasets, confirming the benefits of our learnable model compared to a state-of-the-art perturbation one.

## 5.5 CASE STUDY

Qualitative results are shown in Figure 3. Cases 3a and 3b show examples where adding edges restores the complete motif and class. They illustrate the same input graph, where its motif becomes a cycle (class 0) when one edge is added and a house motif (class 1) when two are added. This verifies that our method can generate multiple counterfactuals for the same input, providing insight into the model's decision boundaries. Out of 23 such cases our reconstruction step successfully recovered 20, while none of the CFExplainer variants produced counterfactuals for class 2 in the incomplete dataset, since this requires edge additions. For instance, case 3c is a typical situation

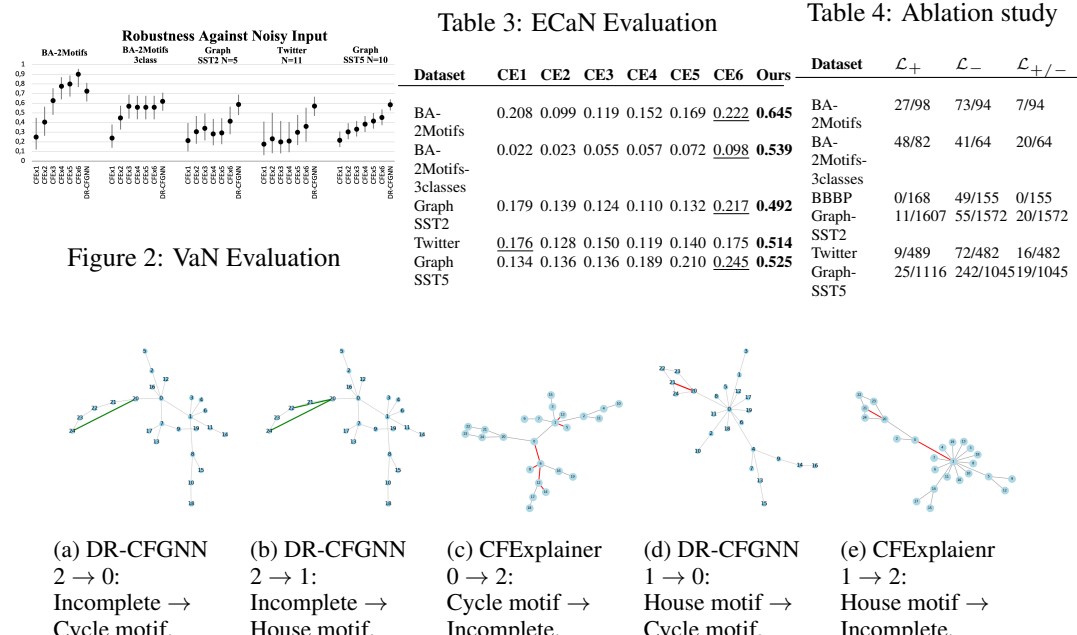

Figure 2: VaN Evaluation

Table 3: ECaN Evaluation

| Dataset | CE1 | CE2 | CE3 | CE4 | CE5 | CE6 | Ours |
|---|---|---|---|---|---|---|---|
| BA-2Motifs | 0.208 | 0.099 | 0.119 | 0.152 | 0.169 | 0.222 | **0.645** |
| BA-2Motifs-3classes | 0.022 | 0.023 | 0.055 | 0.057 | 0.072 | 0.098 | **0.539** |
| Graph SST2 | 0.179 | 0.139 | 0.124 | 0.110 | 0.132 | 0.217 | **0.492** |
| Twitter | 0.176 | 0.128 | 0.150 | 0.119 | 0.140 | 0.175 | **0.514** |
| Graph SST5 | 0.134 | 0.136 | 0.136 | 0.189 | 0.210 | 0.245 | **0.525** |

Table 4: Ablation study

| Dataset | $\mathcal{L}_+$ | $\mathcal{L}_-$ | $\mathcal{L}_{+/-}$ |
|---|---|---|---|
| BA-2Motifs | 27/98 | 73/94 | 7/94 |
| BA-2Motifs-3classes | 48/82 | 41/64 | 20/64 |
| BBBP | 0/168 | 49/155 | 0/155 |
| Graph-SST2 | 11/1607 | 55/1572 | 20/1572 |
| Twitter | 9/489 | 72/482 | 16/482 |
| Graph-SST5 | 25/1116 | 242/1045 | 19/1045 |

(a) DR-CFGNN
2 → 0:
Incomplete →
Cycle motif.

(b) DR-CFGNN
2 → 1:
Incomplete →
House motif.

(c) CFExplainer
0 → 2:
Cycle motif →
Incomplete.

(d) DR-CFGNN
1 → 0:
House motif →
Cycle motif.

(e) CFExplaienr
1 → 2:
House motif →
Incomplete.

Figure 3: CFXs for the BA-2motifs-3classes, showing edge addition (green) and removal (red).

where the removal of edges is evidently not the best choice; the cycle pattern remains, while the graph modification is so radical that does not justify the counterfactual.

Case 3d in Figure 3 is a typical example of a transformation with a single edge removal, tied to the motif, due to the ability of DR-CFGNN to stay close to the factual graph. Notably, there are situations that CFExplainer does not succeed even in cases like this. Case 3e is a similar graph, where, in addition to the cycle creation, a rather valuable edge is also removed, disconnecting the important nodes from the rest without apparent cause. Further results are provided in Appendix A.4.

## 5.6 ABLATION STUDY

Finally, we analyze the contribution of each component of DR-CFGNN for generating counterfactual graphs. Table 4 reports the number of input graphs with at least one CFX, demonstrating that each step contributes under different conditions (the total number in the $2^{nd}$ and $3^{rd}$ columns is less than in the $1^{st}$, since in some cases the factual's graph predictions contradict the input's graph, and are omitted). For example, the reconstruction step ($\mathcal{L}_+$) plays more active role in the incomplete dataset. In contrast, for the molecular dataset, edge addition does not produce CFXs, neither in the reconstruction ($\mathcal{L}_+$) nor in the deconstruction–reconstruction step ($\mathcal{L}_{+/-}$), since real molecular structures cannot generally be modified by simply inserting an additional bond. For the remaining datasets (Graph-SST2, Twitter, and Graph-SST5), whose structure is less clear as they capture text-based relations, the framework is still able to generate a substantial number of counterfactuals.

## 6 CONCLUSION

In this study, we propose a novel framework for local-level counterfactual explainability in GNNs that accounts for any type of edge edits, without performing computationally intensive perturbations. The novelty stems from the architecture design that distinguishes between edge removal and addition learning, enhancing generalizability and efficiency. We evaluate our approach with commonly adopted metrics and propose novel variants to better capture performance. Our method significantly progresses state-of-the-art, and works especially well with incomplete graph data. The ablation study and the overall performance of DR-CFGNN in predicting missing links motivates the need for exploring further the aspect of incompleteness in the context of GNN explainability in the future, which constitutes a pragmatic need for real-world domains.

REPRODUCABILITY

Code for reproducing our experiments is available at an anonymous repository: `https://anonymous.4open.science/r/DR-CFGNN-8396/README.md`

A complete description of the datasets used is given in 5.1. The training hyperpatameters of GNNs and of the DR-CFGNN framework are provided in Appendix A.1.1 and A.1.2, respectively.

In their survey paper, Yuan et al. (2023) developed an open source library for GNN explainability, released as part of the DIG (Dive into Graphs) framework, a turnkey library for graph deep learning research (`https://github.com/divelab/DIG`). We make use of their pretrained graph classification models and the explanation results of the Subgraphx factual explainer. To create the incomplete synthetic dataset or to add noise, we modified the relevant scripts in the DIG library. The comparison method we used, CFExplainer, was also implemented by the authors within the DIG framework.

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

# A APPENDIX

## A.1 IMPLEMENTATION DETAILS

### A.1.1 GNN SETTINGS

We train a 3-layer GCN on all datasets with ReLU activations. We apply a readout function (mean, max, or identity) to aggregate node embeddings from node-level embeddings to graph-level embeddings. Table 5 presents the key hyperparameters and the achieved accuracy for each dataset. Our architecture follows the design by Yuan et al. (2023) for each dataset.

Table 5: Dataset-specific hyperparameters, readout, and achieved accuracy.

| Dataset | Batch Size | Epochs | Readout | Size of layers | Accuracy |
|---|---|---|---|---|---|
| BA-2Motifs | 64 | 800 | mean | 20, 20, 20 | 0.98 |
| BA-2Motifs-3Classes | 64 | 800 | mean | 20, 20, 20 | 0.82 |
| BBBP | 32 | 200 | max | 128, 128, 128 | 0.8195 |
| Graph-SST2 | 128 | 50 | max | 128, 128, 128 | 0.8825 |
| Twitter | 128 | 50 | max | 128, 128, 128 | 0.7066 |
| Graph-SST5 | 128 | 50 | max | 128, 128, 128 | 0.5050 |

### A.1.2 DR-CFGNN SETTINGS

We restrict the size of our explanations to $r + k$, two user-defined hyperparameters, where $r$ is the maximum number of deleted edges and $k$ is the maximum number of added edges. In all our experiments, we set both to 2 to ensure that the explanations remain small and simple. This choice is conceptually similar to the $C$-sparse metric introduced by Liu et al. (2021), where the explanation graph is constrained to contain no more than C nodes.

To further control the quality of counterfactuals, edge additions were restricted to candidate edges with prediction scores above a fixed threshold of 0.8. For each dataset, we allowed up to $k = 2$

sequential edge additions per input graph, where edges were added cumulatively in descending score order, potentially yielding multiple valid counterfactuals.

For factual explanations, the subgraph size was fixed per dataset: 6 for BA-2Motifs, BA-2Motifs-3Classes, and BBBP; half the average number of nodes (as reported in Yuan et al. (2023)) for Graph-SST2/5 and Twitter. Counterfactuals were then generated by masking combinations of up to $r = 2$ edges from these factual subgraphs.

## A.2 EVALUATION METRICS

**Probability of Necessity (PN)**: PN computes the proportion of input graphs for which at least one counterfactual exists, i.e.,

$$
\text{PN} = \frac{1}{M} \sum_{G \in \mathcal{G}} \begin{cases} 1, & \text{if } \exists G^{CF,i} \text{, s.t. } \Phi(G^{CF,i}) \neq \Phi(G_i), \\ 0, & \text{otherwise,} \end{cases} \tag{7}
$$

where $G^{CF,i}$ is some counterfactual for instance $G_i$ and $M = |\mathcal{G}|$ the size of the set of input graphs.

**Explanation Size**: The explanation size is defined as:

$$
ExpSize(G, G^{CF}) = \sum_{i,j=1}^{N} |a_{i,j} - a_{i,j}^{cf}| + nnz(\mathbf{F}) \text{, s.t. } a_{i,j} \in \mathbf{A}, a_{i,j}^{cf} \in \mathbf{A}^{CF}, \tag{8}
$$

where $G = \{\mathbf{A}, \mathbf{X}\}$ and $G^{CF} = \{\mathbf{A}^{CF}, \mathbf{X}^{CF}\}$ the input and the counterfactual graphs respectively, $N$ the size of the graphs, $\mathbf{F}$ the feature mask on $\mathbf{X}$ that results to $\boldsymbol{X}^{CF}$ and $nnz(\cdot)$ indicates the number of non-zero entries in the matrix (see Eq. 2).

**Motif Proximity**: Let $\mathcal{G}^{CF,i}$ the set of counterfactual graphs generated for input graph $G_i \in \mathcal{G}$. Let also $\mathcal{E}_{i,j}^{diff}$ the set of edge modifications (added or removed) occurred to $G_i$ that lead to $G_j^{CF,i}$. The Motif Proximity is defined as

$$
\text{MotifProx} = \frac{1}{M^*} \sum_{i=1}^{M^*} \max_{G_j^{CF,i} \in \mathcal{G}^{CF,i}} \left( \frac{\sum_{e \in \mathcal{E}_{i,j}^{diff}} \begin{cases} 1, \text{ if } e \text{ touches at least one motif node} \\ 0, \text{ otherwise} \end{cases}}{ExpSize(G_i, G_j^{CF,i})} \right), \tag{9}
$$

where $M^*$ is the number of input graphs with at least one counterfactual explanation.

**Validity after Noise**: Given $G_i \in \mathcal{G}$ with counterfactuals $\mathcal{G}^{CF,i}$ and its noisy counterpart $G_i^{\sigma} \in \mathcal{G}$ with counterfactuals $\mathcal{G}_{\sigma}^{CF,i}$, we consider all possible pairs between $\mathcal{G}^{CF,i}$ and $\mathcal{G}_{\sigma}^{CF,i}$ with identical counterfactual predictions $c$. Based on these pairs for each $G_i \in \mathcal{G}$, we compute totals across all input graphs. We denote $x$ the number of input graphs for which at least one corresponding counterfactual graph with the same $c$ still exists after the introduction of noise, and $n\prime$ the number of input graphs that had at least one counterfactual explanation before adding noise. We define:

$$
\hat{p} = \frac{x}{n\prime}. \tag{10}
$$

Here, each distinct counterfactual prediction per input graph is treated individually when computing $x$ and $n\prime$. Thus, we implement VaN with the help of Wilson Score Interval as:

$$
VaN(x, y) = \frac{n\prime}{n\prime + z^2} \left[ \hat{p} + \frac{z^2}{2n\prime} \pm z \sqrt{\frac{\hat{p}(1 - \hat{p})}{n\prime} + \frac{z^2}{4n\prime^2}} \right]. \tag{11}
$$

**Edge Consistency after Noise (ECaN)**: Let $\mathcal{E}_{i,j}^{diff}$ and $\mathcal{E}_{\sigma}{}_{i,j}^{diff}$ the set of edge modifications occurred to $G_i$ that lead to $G_j^{CF,i}$ and $G_i^{\sigma}$ that lead to $G_{\sigma,j}^{CF,i}$, respectively. Then, ECaN is defined as:

$$
\text{ECaN} = \frac{1}{n\prime} \sum_{i=1}^{n\prime} \max_{\substack{j \\ \text{with } \Phi(G_j^{CF,i}) = \Phi(G_{\sigma,j}^{CF,i})}} \frac{|\mathcal{E}_{i,j}^{diff} \cap \mathcal{E}_{\sigma}{}_{i,j}^{diff}}{|\mathcal{E}_{i,j}^{diff} \cup \mathcal{E}_{\sigma}{}_{i,j}^{diff}|}. \tag{12}
$$

## A.3 EXTENDED QUANTITATIVE ANALYSIS

Table 6 shows the scores of the Validity after Noise (Van) along with the corresponding Wilson confidence intervals, which are also depicted in Fig. 2.

Table 6: Validity after Noise (VaN) across datasets and methods, with Wilson confidence intervals.

| Dataset | CFEx1 | CFEx2 | CFEx3 | CFEx4 | CFEx5 | CFEx6 | DR-CFGNN |
|---|---|---|---|---|---|---|---|
| BA-2Motifs | 0.250 (0.120, 0.449) | 0.405 (0.263, 0.565) | 0.628 (0.479, 0.756) | 0.776 (0.641, 0.870) | 0.800 (0.670, 0.888) | **0.900** (0.786, 0.957) | 0.725 (0.619, 0.811) |
| BA-2Motifs-3classes | 0.239 (0.139, 0.379) | 0.448 (0.327, 0.575) | 0.569 (0.441, 0.688) | 0.559 (0.433, 0.678) | 0.559 (0.433, 0.678) | 0.559 (0.433, 0.678) | **0.620** (0.522, 0.709) |
| Graph SST2 | 0.214 (0.102, 0.395) | 0.306 (0.180, 0.469) | 0.341 (0.216, 0.495) | 0.282 (0.165, 0.438) | 0.293 (0.176, 0.445) | 0.415 (0.278, 0.566) | **0.588** (0.478, 0.689) |
| Twitter | 0.176 (0.062, 0.410) | 0.231 (0.082, 0.503) | 0.200 (0.081, 0.416) | 0.208 (0.092, 0.405) | 0.300 (0.167, 0.479) | 0.360 (0.202, 0.555) | **0.571** (0.469, 0.668) |
| Graph SST5 | 0.216 (0.146, 0.308) | 0.304 (0.228, 0.394) | 0.331 (0.255, 0.416) | 0.383 (0.303, 0.469) | 0.416 (0.337, 0.500) | 0.453 (0.373, 0.536) | **0.584** (0.526, 0.641) |

## A.4 EXTENDED QUALITATIVE ANALYSIS

We continue our qualitative analysis with the BA-2Motifs-3Classes synthetic dataset, focusing on counterfactual graphs produced by CFExplainer. Figures 4 to 9 visualize CFs produced by each CF-Explainer variant, respectively. Note that none of these variants generate counterfactual explanations capable of reconstructing the incomplete motif (from class 2 to 0 or 1).

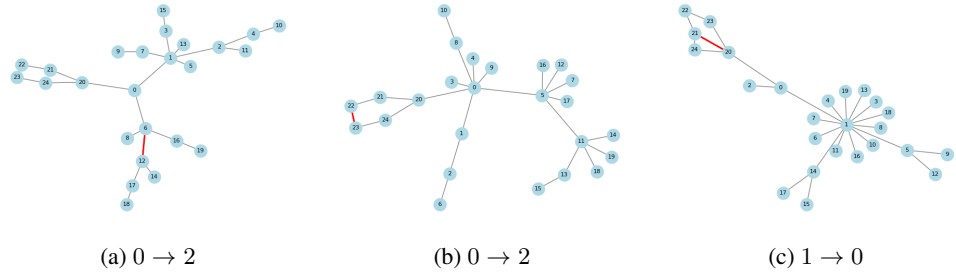

(a) $0 \to 2$         (b) $0 \to 2$         (c) $1 \to 0$

Figure 4: Counterfactual graphs generated by CFEx1. Each subfigure shows input graph prediction $\to$ counterfactual prediction. Red edges indicate deleted edges.

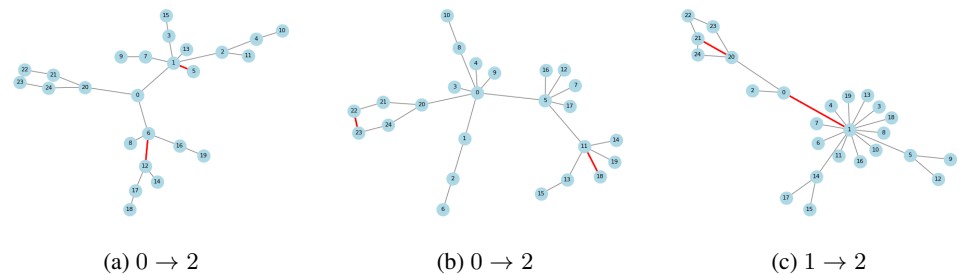

(a) $0 \to 2$         (b) $0 \to 2$         (c) $1 \to 2$

Figure 5: Counterfactual graphs generated by CFEx2. Each subfigure shows input graph prediction $\to$ counterfactual prediction. Red edges indicate deleted edges.

In Figure 4b, an edge inside the motif was deleted from the input graph, and the counterfactual graph predicted the incomplete class, as expected. Similarly, in Fig. 4c, removing the "roof" edge changed the motif from a house to a cycle. However, in Fig. 4a, the prediction changes even though the removed edge is far from the motif. We observed many such cases where the counterfactual explanation does not reflect the underlying motif structure, as indicated by the motif proximity

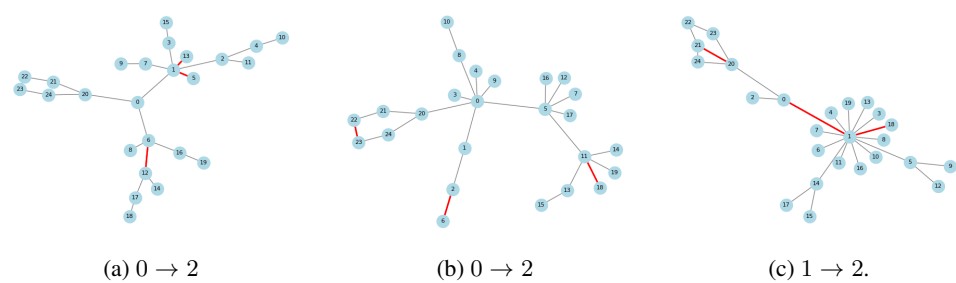

(a) $0 \rightarrow 2$          (b) $0 \rightarrow 2$          (c) $1 \rightarrow 2$.

Figure 6: Counterfactual graphs generated by CFEx3. Each subfigure shows input graph prediction $\rightarrow$ counterfactual prediction. Red edges indicate deleted edges.

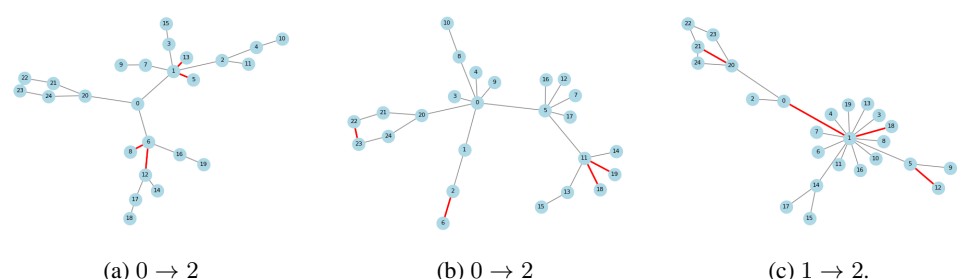

(a) $0 \rightarrow 2$          (b) $0 \rightarrow 2$          (c) $1 \rightarrow 2$.

Figure 7: Counterfactual graphs generated by CFEx4. Each subfigure shows input graph prediction $\rightarrow$ counterfactual prediction. Red edges indicate deleted edges.

metric. Notably, as the explanation size (number of deleted edges) increases from 2 to 6 (Fig. 5a to Fig. 9a), none of the explanations contain even a single edge inside the motif. In general, larger explanations tend to be less meaningful due to the nature of the dataset and the motifs.

Below, we focus on the Deconstruction/Reconstruction step for the BA-2Motifs-3Classes dataset. We selected an input graph where counterfactual graphs were produced using the Deconstruction/Reconstruction step, but not by using only the Reconstruction step. This combined step can also serve as a correction to the SubgraphX factual explainer. Figure 10 shows that the deleted edges alone do not provide a meaningful explanation and do not change the prediction, whereas the addition (Reconstruction) step does lead to a prediction change.

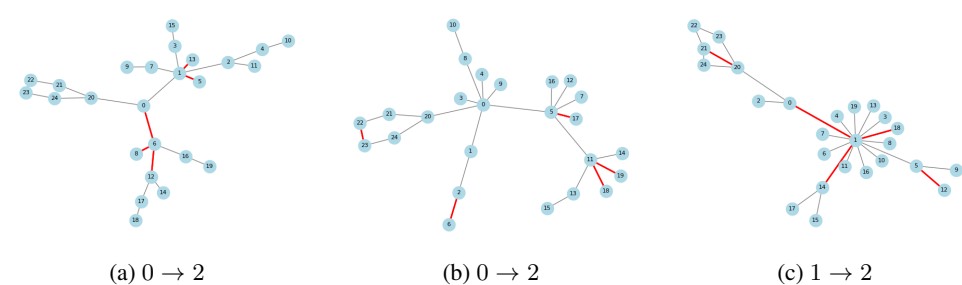

(a) $0 \to 2$     (b) $0 \to 2$     (c) $1 \to 2$

Figure 8: Counterfactual graphs generated by CFEx5. Each subfigure shows input graph prediction → counterfactual prediction. Red edges indicate deleted edges.

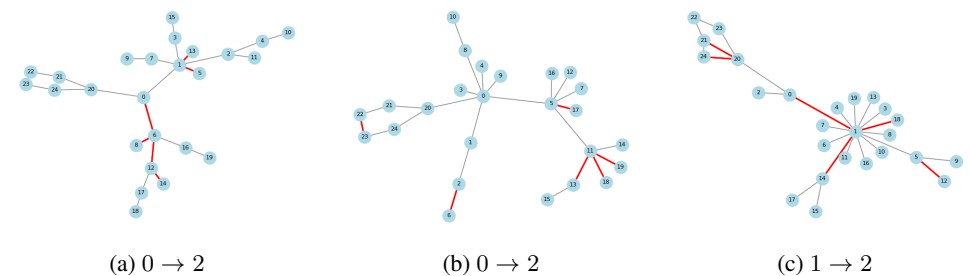

(a) $0 \to 2$     (b) $0 \to 2$     (c) $1 \to 2$

Figure 9: Counterfactual graphs generated by CFEx6. Each subfigure shows input graph prediction → counterfactual prediction. Red edges indicate deleted edges.

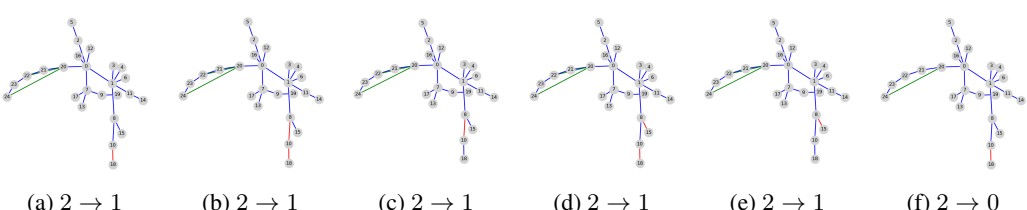

(a) $2 \to 1$   (b) $2 \to 1$   (c) $2 \to 1$   (d) $2 \to 1$   (e) $2 \to 1$   (f) $2 \to 0$

Figure 10: Counterfactual graphs from the Deconstruction/Reconstruction step; red edges are deletions and green edges are additions.

