# OpenReview forum: "DR-CFGNN: A Completion-Aware Framework for Counterfactual Explainability in Graph Neural Networks"
_ICLR.cc/2026/Conference — ICLR 2026 Conference Withdrawn Submission_

### Official Review · Reviewer_2PZj · 2025-10-20

**Soundness:** 3
**Presentation:** 3
**Contribution:** 2
**Rating:** 4
**Confidence:** 2

**Summary:**

This paper proposes a generic, model-agnostic method for local-level GNN explainability that considers both edge removal and edge assertion. The approach takes advantage of the advanced explanation objectives through an ensemble design.
Experiments show that the proposed method not only achieves state-of-the-art performance on standard benchmark datasets but also generates more robust explanations when data incompleteness occurs.

**Strengths:**

- This paper proposes a novel ensemble framework for counterfactual explainability in graph neural networks (GNNs).
- The study of the robustness of counterfactual explanations based on noise is interesting.
- The paper overall is easy to follow.

**Weaknesses:**

1. The method proposed in the paper resembles an ensemble explainer, as it integrates candidate explanations from multiple explanatory objectives. Reporting the best-performing one among these will always yield results no worse than the best explanations from a single objective, and this approach can intuitively improve performance across metrics. However, each of these objectives can be found in existing works such as [1], so the contribution to explanatory methods is not strong.

2. Regarding the paper’s contributions to the evaluation system, they mainly come from the integration of existing metrics. Although the authors are the first to evaluate the impact of counterfactuals in dealing with incomplete graph data through a noisy-based metric, the relevant theories and implementation methodologies also originate from existing works such as [2]. As one of the key contributions claimed in the paper, this noisy-based metric lacks discussion and analysis on the necessity of its introduction for GNN explanation. This necessity determines whether it can be adopted as a new general metric for subsequent GNN explanation works.

3. Eq. (3), Eq. (4), and Eq. (5) are confusing. The right-hand side of the equal sign represents the edited adjacency matrix, yet the graph label does not appear in the equations. I am confused about how it optimizes the objective described in the context.

In summary, when considering the paper’s contributions to explanatory methods and evaluation methods in isolation, they are not strong. Therefore, it is more necessary to supplement analyses that previous works have not conducted, such as discussing the necessity of introducing the new metric, and explaining why an ensemble is necessary and why a unified objective cannot be used to extract explanations in an end-to-end manner. However, the paper lacks such analyses and fails to provide new impactful insights.

- [1] Joint factual and counterfactual explanations for top-k gnn-based recommendations.
- [2] Robust counterfactual explanations in machine learning: a survey.

**Questions:**

Please refer to the above weakness section for suggestions and questions.

---

> ### Author Response · Authors · 2025-11-21
>
> The authors sincerely thank the reviewer for their helpful and constructive feedback. The comments are very valuable in helping us to better articulate the novelty of our method and clarify our motivations.
>
> Regarding weakness 1: we would like to clarify that our method should not be seen simply as an ensemble explainer. The three steps of our approach are designed to address different counterfactual tasks-removing influential structure, adding potentially missing edges or applying both-and each is used when appropriate. Typically, ensemble methods apply different models to solve the same tasks; in our case, we argue in the paper that the subtasks should be seen as of different nature, therefore each should be approached with a dedicated model.These steps also highlight that a single objective is often not sufficient to generate a meaningful counterfactual explanation.
>
> Regarding weakness 2: While previous works such as [2] discuss the usefulness of studying robustness after adding noise to the input, we formulate two metrics mathematically. Our method generates multiple counterfactuals, and we define a counterfactual as robust if at least one exists after the addition of noise. Incorporating the Wilson score further improves the statistical reliability of this evaluation. To the best of our knowledge, we are also the first to propose Jaccard similarity as a metric for the robustness of counterfactual explanations under noise. At the same time, we acknowledge that our discussion of the proposed noisy-based metric (ECaN) could have been more thorough, and a deeper analysis of its necessity and impact would strengthen the argument for its adoption as a general evaluation tool.
>
> Regarding weakness 3, indeed matrices M and F are implicitly passed as arguments to the loss functions; we thank the Reviewer for observing this, we will revise the formulas accordingly.

---

> > ### Comment · Reviewer_2PZj · 2025-11-25
> >
> > Did you include the new result about noisy-based metric and correct the formula in the revision? Please highlight the newly added results in the revision (maybe using red colored text).

---

> > > ### Author Response · Authors · 2025-11-28
> > >
> > > We are indeed working on an improved version in line with the suggestions given. However, we decided not to upload a revised version at this stage, as incorporating the additional material suggested by all reviewers would require substantial changes beyond the scope of the current submission. We again want to thank the reviewer for their thoughtful feedback.

---

### Official Review · Reviewer_ZXSc · 2025-10-28

**Soundness:** 2
**Presentation:** 2
**Contribution:** 2
**Rating:** 4
**Confidence:** 3

**Summary:**

The paper proposes a method for constructing counterfactual graphs through edge addition and edge removal, leveraging a graph classification model and a factual graph explainer.

To identify minimal edge removals, the method first extracts a subgraph that preserves the original classification result using a factual graph explainer, then exhaustively searches candidate subgraphs by removing edges from this factual subgraph to disrupt the pattern.

For edge addition, the approach classifies graphs with incomplete motifs and those with complete motifs into two categories, and trains a GNN to predict the edges required to complete the motif so that the graph’s class label changes accordingly.

The main contribution of this paper lies in exploring the potential of decoupling edge addition and edge removal, and in analyzing their respective roles in generating counterfactual examples.

**Strengths:**

1. The paper proposes a new framework for identifying counterfactual examples in post-hoc graph neural network (GNN) analysis.

2. It explores the potential of separating edge addition and edge removal, and analyzes their respective effects on discovering counterfactual examples.

3. The proposed model demonstrates advantages in specific scenarios, such as recognizing graph incompleteness.

**Weaknesses:**

1. The paper lacks methodological novelty. The edge removal component essentially performs an exhaustive search based on the results of existing factual GNN explainers, while the edge addition component closely resembles traditional link prediction methods. Overall, the contribution is insufficient to meet the novelty standards expected for ICLR.

2. Several ambiguous descriptions appear in key sections, making the paper difficult to follow. More detailed comments are provided in the following section.

3. The reported performance improvement over CFX primarily arises from the experimental setting involving incomplete motifs, which is specifically tailored to favor the proposed edge addition approach. Therefore, this advantage cannot be considered a general one.

**Questions:**

In Definition 3.1, the graph is inconsistently represented as ${A, X}$ and ${M, F}$. Please clarify whether these notations represent different formulations.
In Section 4.1, the loss function is expressed as a matrix, which is an unconventional formulation in machine learning. It would be helpful to include an explanation or derivation of how this matrix loss is computed.
In Equation (6), $L_{+/-}$ appears to contain the same term as $L_{+}$ or $L_{-}$. Readers may find it confusing why these terms need to appear again in the final loss formulation without a clarification addressing this overlap.
The feature mask introduced in the edge-removal method and the feature noise used in the experiments do not seem directly related to the proposed approaches. Please clarify their roles and how they contribute to the main methodology.
In Section 5.3, it is unclear why $r = k = 0$ is considered a trivial case. Although the total number of edges remains unchanged, different edge combinations could still alter or disrupt structural patterns in the graph.
In Table 1, no explanation size is reported for the CFE models. Since explanation size and PN are typically considered together in a trade-off, including explanation size would allow for a fairer and more complete comparison.
In RQ3, the paper notes that adding noise could result in invalid molecular graphs, yet similar concerns are not raised for textual or semantic noise in sentimental analysis. Please clarify why these two types of noise are treated differently.

---

> ### Author Response · Authors · 2025-11-21
>
> We wish to thank Reviewer ZXSc for the detailed examination of our work and for pointing out aspects that can help us improve our contribution.
>
> Regarding Definition 3.1, it is a leftover, we agree that $G^{CF} = \{M,F\}$ should be corrected to $G^{CG}= \{A’,X’\}$.
>
> Regarding Section 4.1 and the formulation of the loss function, we can understand that this may seem unconventional to the broader Machine Learning community. In the context of GNN explainability, there is a notable lack of coherency in how to represent and how to evaluate the different models. To the best of our knowledge, the recent study by Guo et al. (2025) is the first to make a systematic attempt to unify 13 different explainers into a general framework (Section 4.2 in that paper). In that approach, the problem is cast into a task of optimizing an objective function that contains matrices. We chose not to deviate from that perspective, in order to make our proposed framework easier to compare with existing systems (see also our response to Reviewer R19j). Nevertheless, we appreciate the remark that this may make the formalization less comprehensive to the broader community; more clarifications will be introduced.
>
> Regarding the comment on Eq. 6: without this seemingly, yet necessary, redundancy, we can optimize explanation generation based either on edge removal or on edge addition, but not both. On the other hand, if we only introduced the L+/- loss, we face the risk of confusing that this is an end-to-end optimization process, which contradicts one of the main points of our study. The explanation provided briefly below Eq 5 in the paper can be improved to avoid confusion.
>
> We thank the reviewer for the remark regarding the role of node features and their handling in our framework, which prompts us to expand further the properties of our approach. Node features play a significant role in many real-world graph data and the reconstruction step presented in Section 4.2 is designed to generate embeddings considering both a node’s neighborhood and the corresponding node features. In this respect, our framework is able to learn to identify important features, but more importantly to be robust to noise, as the experiments showed. Regarding feature masking though, although such an implementation is not included in our model (yet), it is clear that a similar (but more efficient) process as the one described in section 4.1 for generating subgraphs, given the factual graph, can be performed to generate subsets of features, once the unimportant features are ironed out by the factual explainer in the previous step.
>
> Regarding the comment concerning the r=k=0 case, we would like to clarify that different edge combinations, even when they do not change the original number of edges of the graph, are calculated as changes. The r=k=0 case is trivial, as it corresponds to leaving the factual graph that is returned in the destruction step intact, which by definition does not produce any counterfactuals.
>
> Regarding Table 1 : for CFExplainer, the explanation size must be fixed before running the method. This explanation size is directly encoded in the method name (e.g., CFEx1 has size 1, CFEx2 has size 2, … CFEx6 has size 6), which is why these sizes are not repeated in parentheses.
>
> Regarding RQ3: According to the rationale discussed by Jiang et al. (2024a) and in other similar studies, in order to assess robustness, the noise introduced needs to be small enough so that it does not substantially modify the input, therefore not justifying a change to the target GNN's prediction. For a molecule dataset, even small structural modifications may cause significant changes in the properties of molecules; lacking ground truth, it is difficult to determine where to introduce noise in an appropriate manner.

---

### Official Review · Reviewer_FeuL · 2025-10-30

**Soundness:** 2
**Presentation:** 2
**Contribution:** 2
**Rating:** 2
**Confidence:** 5

**Summary:**

The paper proposes a framework for local counterfactual explainability that jointly considers edge removals and additions. The experiments indicate that the proposed method outperforms the baselines.

**Strengths:**

- The paper is well-motivated, and the problem is clearly defined.
- The method is explained clearly.

**Weaknesses:**

- The presentation of tables and figures is difficult to read; some tables are very small and not visually clear. The overall visual presentation needs improvement.
- Novelty appears limited; the paper does not convincingly position its methodological contribution.
- The number of baselines is limited, focusing on only one baseline and its variants.
- The algorithm’s computational complexity is not discussed, and running times are not reported.
- The graphs used in experiments are small; please provide, preferably in a table (Appendix is fine), the average number of nodes and edges for each dataset.
- The shared code includes a requirements.txt that references local paths and lacks version specifications, which hinders reproducibility.

**Questions:**

- A random algorithm outperforms the main algorithm on one of the most important metrics. Can you elaborate on why this occurs? Is it due to small graph sizes? If so, why should one prefer the ML-based approach over a random baseline?
- Can you provide a general requirements.txt with pinned package versions and the Python version? Also, please share details of your runtime environment (e.g., CPU/GPU, memory, and machine specifications) to support reproducibility.
- Can you report the algorithm’s time complexity and empirical running times?

Additional suggestions for readability and improvements:
- Please refer explicitly to the table for RQ4 in Section 5.4.
- Consider enlarging tables, adopting consistent formatting, and adding visual cues (e.g., row/column grouping, margins, fonts) to improve readability.
- Strengthen the novelty narrative by clearly contrasting your approach with existing methods and articulating the specific scenarios where your framework offers unique advantages.
Expand the baseline set to include diverse and stronger comparators to better contextualize performance.

---

> ### Author Response · Authors · 2025-11-21
>
> We would like to thank Reviewer FeuL30 for the time devoted to review our paper and for the insightful suggestions, including the careful examination of the code. We understand the concerns raised regarding the number of baselines, novelty and running times. To include such additional material in this version would require substantial changes to the other sections, in order to preserve the paper's coherence; we will sincerely consider these comments for an improved version of our work.
>
> We would only like to respond to the Reviewers question regarding the size of the graphs and why the random baseline outperforms the other models in the PN metric, to help clarify any misconceptions.
>
> Regarding Question 1, each of the synthetic datasets contains 25 nodes, while the Graph-SST2, Graph-SST5, and Twitter graphs have on average 10.199, 19.848, and 21.103 nodes, respectively. Given these relatively small graph sizes and the exhaustive search performed by the random baseline within a 60-second limit per graph, it is able to achieve high coverage, which boosts its PN score.
>
> We would also like to point out that, while indeed PN is widely adopted in relevant literature as a key metric, its interpretation can often be misleading; as we show with our experiments, finding at least some counterfactual (which is what the PN metric aims to capture) should not be considered a satisfactory solution without further exploration. A critical parameter to our opinion is the quality of the explanation. In order to assess quality, metrics capturing robustness or motif proximity can provide valuable insights. In this respect, algorithms relying on random-based algorithms seem to face difficulties converging to explanations that satisfy quality criteria.

---

### Official Review · Reviewer_R19j · 2025-10-31

**Soundness:** 1
**Presentation:** 1
**Contribution:** 2
**Rating:** 2
**Confidence:** 4

**Summary:**

This paper proposes a counterfactual explanation framework for graph neural networks that aims to identify important subgraphs and provide interpretability through a decomposition-and-masking procedure. While the motivation is sound, the paper suffers from weak novelty justification, and insufficient experimental rigor. Moreover, several key methodological and presentation issues significantly hinder readability and the evaluation of contributions.

**Strengths:**

1.	The paper addresses the important topic of counterfactual explanations in graph models, which is relevant and timely.
2.	The experimental section provides some visualization and ablation, which indicates implementation effort.

**Weaknesses:**

1. Weak and Redundant Challenge Definition
The listed “challenges” are not convincing.
*	The first two challenges both focus on subgraph reasoning or graph expansion, which have already been extensively explored in prior works such as GCFExplainer. Therefore, the motivation for re-stating these as novel challenges is weak.
*	Furthermore, the paper does not analyze the limitations of existing “graph expansion” methods (e.g., GCFExplainer), nor explain what unique challenge this work addresses beyond them.
*	The fourth challenge mentions that “most models implement costly search methods,” yet the proposed paper provides no discussion or quantitative analysis regarding computational complexity or runtime cost, which makes this challenge invalid.
2. Problem Formulation Unclear:  The problem formulation section is confusing and not well aligned with existing definitions of counterfactual explanation. The authors cite Guo et al. and CFExplainer (the baseline), but these works already provide a formal and widely accepted definition of the counterfactual explanation task on graphs. Since the current paper does not define a new problem, the formulation should directly follow this established setting instead of introducing vague “decomposition” and “mask” operations, which belong to the method design, not the problem formulation.
3. Unclear Method Innovation: Due to the challenges are vague and overlap with existing work, it is difficult to understand the design motivation of the proposed framework. The method seems to combine existing decomposition and masking steps without a clear theoretical or algorithmic novelty. The paper does not demonstrate why these design choices are necessary or how they overcome specific weaknesses of prior methods.
4. Baseline Selection Insufficient: The experiments only compare with CFExplainer, which is inadequate. At minimum, the authors should include Grad-CAM, or GCFExplainer, the relative methods mentioned in your paper or the paper of CFExplainer, as additional baselines. Without these comparisons, the claimed improvement lacks credibility.
5. Missing Discussion of Hyperparameters: Key hyperparameter choices are not justified. For example, in RQ3, the perturbation ratio is fixed to 4%, but the paper provides no explanation for this choice, nor any sensitivity analysis. Such details are essential to assess the robustness and fairness of the method.
6. Experimental Presentation and Formatting Issues:
The presentation quality is poor and significantly affects readability:
*	Several figures (e.g., Figure 2) contain multiple subplots in a single image, making the contents unreadable due to small font and low resolution.
*	Table formatting is inconsistent across the paper, with misaligned columns and missing captions.
*	Figure labels and axis annotations are too small to interpret.
*	Overall, the visual quality and layout do not meet publication standards.

**Questions:**

Please address the weaknesses.

---

> ### Author Response · Authors · 2025-11-21
>
> We are grateful to Reviewer R19j for carefully reading our paper and for providing comments, which we will take seriously into consideration. Although we understand and agree with most of the points raised by the Reviewer, we feel that some clarifications are necessary, not so much in order to argue on the overall assessment, but rather to ensure that certain impressions resulting from the review accurately reflect our research effort.
>
> Regarding the remark on the problem formulation, which does not align well with existing definitions. In section 2, we clearly state that we stay close to the formalization of the problem introduced by Guo et al. (2025). Indeed, Eq. 1 in our paper exactly follows Eq. 32 in that study, including notation. A similar practice is followed throughout that section for the definitions of the objective functions, with the necessary simplifications due to space limitations introduced to a conference paper.
>
> This problem definition should not be confused with the methodology, though (see for instance how subsection 4.2.2 in Guo et al. aims to map those definitions to the various methodologies). To avoid any misconception, we keep the problem formulation separate from Section 4 “Our proposed framework..”. As a result, we believe it is clear that we did not intent to define a new problem formulation nor did we overlook existing practice; instead, our goal was exactly to keep the problem definition clear by adopting a formulation that aims to unify diverse methodologies that exist in relevant literature. Please note that, to the best of our knowledge, no established setting exists in the field of GNN explainability, and this is a need raised by many recent studies; we hoped our formulation of the problem could be seen as an effort to work towards that direction, not the other way around.
>
>
> Regarding  the lack of analyzing the limitations of existing graph expansion methods, such as GCFExplainer: we respectfully disagree with the Reviewer's opinion that this direction has been extensively explored. In Section 1, we clearly state that graph expansion has indeed been considered in relevant literature; yet, when focusing on local-level approaches, we notice significant limitations, which we analyse for each individual system separately. GCFExplainer on the other hand is a global-level, random-walks based model. In Sections 1 and 2, we review this and other similar systems, and give our point of view regarding why these two characteristics (global view and random walks) may hinder its applicability to counterfactual explainability tasks. We will be more cautious in a future version of our paper to elaborate more extensively on such limitations and the challenges involved, including also experimental evidence, as the Reviewer suggested.
>
> Regarding the missing discussion of hyperparameters: please note that implementation details are given in Appendix A.1.1. We thank the Reviewer for suggesting additional aspects to be included, in order to help complete the picture.
>
> Overall, we believe that the Reviewer's comments are fair, as are those of the other reviewers, and we appreciate the effort devoted. We understand that some of the concerns are more critical, e.g., regarding the number of baselines, a better elaboration of the novelty introduced, and running times. To include such additional material in this version would require substantial changes to the other sections, in order to preserve the paper's coherence; we will sincerely consider these comments for an improved future version of our work.

---

### Note · Authors · 2025-12-22

I have read and agree with the venue's withdrawal policy on behalf of myself and my co-authors.